# Movement Sonification Techniques to Improve Balance in Parkinson’s Disease: A Pilot Randomized Controlled Trial

**DOI:** 10.3390/brainsci13111586

**Published:** 2023-11-12

**Authors:** Alfredo Raglio, Beatrice De Maria, Monica Parati, Andrea Giglietti, Stefano Premoli, Stefano Salvaderi, Daniele Molteni, Simona Ferrante, Laura Adelaide Dalla Vecchia

**Affiliations:** 1Istituti Clinici Scientifici Maugeri IRCCS, 27100 Pavia, Italy; daniele.molteni@gmail.com; 2Istituti Clinici Scientifici Maugeri IRCCS, 20138 Milan, Italy; beatrice.demaria@icsmaugeri.it (B.D.M.); monica.parati@icsmaugeri.it (M.P.); laura.dallavecchia@icsmaugeri.it (L.A.D.V.); 3Istituti Clinici Scientifici Maugeri IRCCS, 20851 Lissone, Italy; andrea.giglietti@icsmaugeri.it (A.G.); stefano.premoli@icsmaugeri.it (S.P.); stefano.salvaderi@icsmaugeri.it (S.S.); 4Department of Electronics, Information and Bioengineering, Politecnico di Milano, 20133 Milan, Italy; simona.ferrante@polimi.it

**Keywords:** Parkinson’s disease, neurological rehabilitation, gait, postural balance, sonification, music therapy

## Abstract

Background: Movement sonification has been recently introduced into the field of neuromotor rehabilitation alongside Neurologic Music Therapy and music-based interventions. This study introduces the use of musical auditory cues encompassing the melodic-harmonic aspect of music. Methods: Nineteen patients with Parkinson’s disease were randomly assigned to the experimental (*n* = 10) and control (*n* = 9) groups and underwent thrice-weekly sessions of the same gait training program, with or without sonification. Functional and motor parameters, as well as fatigue, quality of life, and the impact of intervention on patients’ well-being, were assessed at baseline (PRE), the end of treatment (POST), and at follow-up (FU). Between-group differences were assessed for each outcome measure using linear mixed-effects models. The outcome measure was entered as the dependent variable, group and time as fixed effects, and time by group as the interaction effect. Results: Mini BESTest and Dynamic Gait Index scores significantly improved in the experimental group (*p* = 0.01 and *p* = 0.03, respectively) from PRE to FU, demonstrating a significant impact of the sonification treatment on balance. No other significant differences were observed in the outcome measures. Conclusions: Larger sample sizes are needed to confirm the effectiveness of sonification approaches in Parkinson’s disease, as well as in other neurological disorders.

## 1. Introduction

Music therapy is extensively utilized in relational and rehabilitative settings, leveraging sound to engage limbic and paralimbic regions, as well as cortical brain areas, such as the motor cortex, supplementary motor area, cerebellum, basal ganglia, and more [1,2]. Due to these connections, music can be regarded as a valuable intervention for neuromotor rehabilitation [3,4]. Rehabilitative potential can be observed both in movement disorders and in associated aspects related to mood and music engagement ability [5]. Musical stimulus is a complexly structured material capable of compensating for lacking motor systems while also engaging pleasure, motivation, and reward systems, thereby keeping the patient more engaged during the training process [1].

Indeed, music therapy has the potential to induce plastic changes across the lifespan, from childhood to the elderly [1,2]. 

These changes involve both motor and auditory sensory-motor areas within the brain [6,7,8,9], facilitated by enhanced connectivity between brain regions brought about by auditory stimuli. As suggested by Schlaug [1], the plastic changes induced by music at key points within the cerebral network can lead to effects that persist even beyond the duration of the rehabilitation training. 

Also, music engenders emotional involvement during the rehabilitation process and establishes a robust motivational foundation, further reinforced through the coupling of auditory stimulus with sensory-motor components [6,7].

The impact of music is well-documented in stroke rehabilitation, as it can enhance various gait parameters, including velocity, cadence, stride length, and balance [10,11,12,13,14,15], as well as upper limb movements [16,17,18,19,20,21], language [22,23,24], mood, and psychological aspects [25,26,27,28,29,30,31]. Gait rehabilitation studies for Parkinson’s Disease (PD) [32,33,34,35] and multiple sclerosis [24,36,37,38] yield comparable results.

Between music-based approaches, Neurologic Music Therapy (NMT) [3] is defined as an evidence-based and standardized practice aimed at addressing sensory, cognitive, and motor deficits resulting from neurological pathologies.

In other music-supported training, movements are associated with specific musical cues that can provide constant feedback on the quality of movement in terms of kinesthetic characteristics. This sound–movement correlation enables the establishment of a virtuous feedback-feedforward loop that supports the patient in performing the gesture in the most accurate and functional way for rehabilitation purposes.

In this direction, recent literature also reports music-supported rehabilitation experiences with “sonification” techniques. In general, sonification is a method that uses sound to represent and convey information that is typically non-auditory in nature [39,40]. In rehabilitation, sonification involves the conversion of data or information concerning movements (in our case, related to the warm-up and gait training) into audible sounds. 

Here is an overview of the process.

The first step in the sonification process is to collect relevant data related to the task they are working on (especially movement patterns and natural rhythm at baseline). The collected data are then mapped to sound parameters. 

This mapping is designed to create a meaningful and intuitive relationship between the data and the resulting sounds. For example, specific movements or changes in a patient’s physiological metrics can be mapped to changes in pitch, volume, or rhythm. Once the data are mapped, sound is proposed as pre-recorded or generated in real-time or to represent the data changes (in the case of this study, in both conditions). These generated sounds may be in the form of melodies, chords, tones, rhythms, etc., depending on the specific application and the goals of the rehabilitation program.

The generated sounds are used to provide feedback and guidance to the individual undergoing rehabilitation. Patients can hear their own movements or physiological changes translated into sound, which can help them better understand and control their bodies or progress in their rehabilitation exercises.

Sonification is an effective way to facilitate training and monitor progress. Patients can use auditory feedback to adjust and improve movements in real-time, as well as to track their progress over time.

Also, sonification can be customized to suit the individual needs and preferences of each patient. This includes adjusting the sound parameters, mapping, and feedback to best align with the rehabilitation goals and the patient’s capabilities.

This type of rehabilitation can also be used for research purposes, allowing healthcare professionals to gather data on patient performance and assess the effectiveness of specific rehabilitation protocols.

Sonification in rehabilitation harnesses the power of sound to translate data and information into a format that is accessible and informative to patients. It enhances the therapeutic process by providing real-time feedback, helping individuals better understand their progress, and promoting more effective rehabilitation outcomes.

Thus, sonification is the process of representing non-auditory data or information in the form of audible signals or musical sounds; it is often used in various scientific, artistic, and communication contexts, allowing people to better understand complex concepts through auditory perception. 

Recent investigations have centered around the “sonification” techniques that can be considered as a musical representation of movement wherein sonorous-music stimuli are paired with patient movements’ mapping. In these scenarios, patient movements are captured by specialized sensors responsible for mapping them and translating them into numerical data. These data are then interpreted by specific software to generate real-time sonification. This procedure allows the patient to ‘audibly perceive’ their own movement, making corrections that align with the goals of the rehabilitation treatment.

Auditory-motor feedback can effectively substitute damaged proprioceptive circuits [41,42,43], thereby enhancing the rehabilitation process. Interventions involving “sonification” facilitate sensorimotor learning, proprioception, and movement planning and execution [44,45], leading to overall improvements in motor parameters. While “sonification”-related studies predominantly focus on upper limb rehabilitation [41,42,46,47], only a limited number pertains to lower limb rehabilitation [48,49,50,51]. 

Notably, this study introduces the use of musical auditory cues encompassing the melodic-harmonic aspect of music. This form of sonification renders the feedback enjoyable, predictable, and potentially effective. Our aim is to use musical stimuli specifically designed to emphasize the required motor exercise gesture of the patient. In this way, we leverage the mechanisms of NMT techniques combined with the neurophysiological rationale underlying neuromotor rehabilitation.

Our proposal involves applying this distinctive sonification approach to gait training and other secondary outcomes (such as perceived fatigue levels, quality of life, and global perceived effects) among neurological populations. In particular, this article focuses on preliminary data regarding a group of PD patients. 

## 2. Materials and Methods

### 2.1. Study Design

This study is part of a single-blinded parallel-group multicenter randomized controlled trial involving patients with stroke, Parkinson’s disease, and multiple sclerosis. It presents preliminary data concerning a sub-group of clinically stabilized PD patients (*n* = 19). 

The study received approval from the Ethics Committee of the Istituti Clinici Scientifici Maugeri IRCCS SpA SB (2419 CE, 23 April 2020) and was registered ClinicalTrial.gov (NCT04876339). All participating patients provided written informed consent before recruitment in accordance with study requirements and data processing regulations.

Consecutive recruitment of PD patients was conducted at the Istituti Clinici Scientifici Maugeri in Lissone (Monza Brianza, Italy). 

Participants were randomized to one of the two intervention programs using a list of codes generated through a permuted-block randomization procedure before the beginning of the study. The list was generated by an automatic assignment system developed in MATLAB (MathWorks, Natick, MA, USA, Version R2020c), and directly managed by the principal investigator of the study.

### 2.2. Inclusion Criteria


Idiopathic PDAge < 80 yearsMini-Mental State Examination (MMSE) > 24Unified Parkinson’s Disease Rating Scale score (UPDRS, Part III) < 28Stabilized disease and drug therapyAltered gait patternsMotor independence during walking (without orthotic devices and aids) but with a pathological pattern.


### 2.3. Exclusion Criteria


Previous or concurrent diseases disabling lower limb functionsChanges in drug therapy during the studyRehabilitative treatments involving music in the year before the study.


### 2.4. Interventions

The experimental group (*n* = 10) underwent a gait training program supported by sonification techniques over 20 sessions (30 min each, three times a week). The control group (*n* = 9) underwent the same gait training program without sonification support (Appendix A). 

The study followed customary clinical practice procedures and adhered to guidelines related to gait rehabilitation.

(a) Gait training program without sonification

The protocol consists of two phases. In the first phase (15 min), the patient performs warm-up therapeutic exercises for gait re-education by dividing the walking motor sequence into different phases and progressively re-educating them. This technique allows for focused training of individual steps.
Anteroposterior load shift in tandem position with left foot forward (3 min)Anteroposterior load shift in tandem position with right foot forward (3 min)Left foot swing (3 min)Right foot swing (3 min)March in place (3 min).

During the second phase (15 min), the patient performs a 14 min walk with a 1 min break in the middle (7 min of walking, 1 min rest, 7 min of walking). In the latter part of the walk, the patient is asked to slightly increase the pace of their steps to the maximum possible speed. 

(b) Gait training program with movement sonification

Gait training program with sonification involves the same exercises described above but incorporates a musical component. Each exercise is supported by pre-recorded musical stimuli that are subsequently produced in real-time with the mediation of a sensor (details below). 

The pace of the steps is measured (the measure is obtained considering a click every half step) and utilized for warm-up exercises in the training’s initial 15 min and for the first 7 min of the subsequent phase. In the final 7 min, after a 1 min break, the patient is prompted to slightly increase their step pace to the maximum achievable speed.

The sonification system comprises two inertial measurement units (IMU) (Xsens, Xsens Technologies B.V., Enschede, The Netherlands) with a sampling frequency of 100 Hz, a PC running Windows 10, and a pair of Bluetooth headphones connected to the computer. The sensors are placed, one per leg, at the ankle during the exercises. The earphones are synchronized with the PC and worn by the patient. Ad-hoc customized software using Matlab program (MathWorks, Natick, MA, USA) records heel-ground contacts from IMU-derived angles and triggers musical stimuli related to heel-ground contact position, audible through the headphones. Sequential steps build a musical progression, regular and predictable, based on correct step sequencing.

The dedicated software manages the demographic and clinical data of the participants, records IMU signals, displays the real-time cadence, and triggers musical stimuli.

In detail, before each exercise, the sensors need to be calibrated. These sensors utilize accelerometers and gyroscopes to measure acceleration and angular velocity in three dimensions, enabling the precise determination of the knee joint angle. They are lightweight, wireless devices that can be worn directly on the body, allowing patients to move freely and naturally.

The data from the sensors are subsequently processed by software specifically developed in MATLAB. This software interprets data on the instant angular positions of the knee joint, deducing the foot strike (the moment when the foot touches the ground) and the cadence (the number of steps per minute).

In the subsequent phase, called “sonification”, a predefined sound sample is played at specific moments of the movement. The foot strike represents the main triggering element. Depending on the specific exercise, a fragment of a sound sequence is played at each foot strike or during the oscillation phase during knee flexion. Musically, sonification actively supports the movement, creating a correspondence as close and connotative as possible. For example, in the walking exercise, sonification matches the foot strikes with the downbeats of the musical sequence, integrating an upbeat sound during the second phase of knee flexion (the upbeat sound corresponds to the highest point of the knee in the walking cycle) (Appendix A).

In addition to the real-time management described, the system can store historical data related to the patient’s walking, allowing for post-training analysis. 

For each first five 3 min warm-up exercises, the first part (1′ 30″) is supported by a chord progression with a click on each musical beat. This cadence guides exercise speed. In the second part of the exercise (1′ 30″), the patient recreates the chord progression listened to in the first part, with each foot contact triggering a chord.

The same methodology is applied during walking. For the first 7 min, a chord progression with clicks guides the patient’s steps. The subsequent 7 min see the patient reproducing the chord progression in real-time, without clicks, associating each foot contact with a chord. The patient is asked to slightly increase the step pace.

Musical accompaniment is structured around two principles: (a) “entrainment” guides patients through rhythm in the initial phase, and (b) “real-time sonification” enhances proprioception once the rhythm is internalized, emphasizing support perception through associated sound (chord progression).

Rhythmic aspects encompass harmonic progressions featuring four-beat movements and an upbeat on the third movement. Different progressions include timbre variations (piano, harp, zither, lute, koto) and varied chord sequences (e.g., I IV I V, I II V I, I VI II V, I IV I V) (Appendix A).

### 2.5. Assessment

Participants were evaluated at baseline (PRE), after 20 sessions (POST, end of treatment), and at follow-up (FU, one-month post-treatment) during the “on” phase of their anti-PD medication. Assessment scales included:

Functional Evaluation:Functional Independence Measure (FIM) [52] also separating the FIM cognitive and FIM motor scores to assess and document the functional status of patients.

Motor Parameter Evaluation:Six-Minute Walking Test [53] to evaluate patient’s exercise capacity and functional endurance;Mini BESTest [53] to assess balance, postural control, and functional mobility; Dynamic Gait Index [53] to evaluate patient’s ability to perform complex walking tasks and assess their dynamic balance during walking;Timed Up and Go [53] to assess patient’s mobility, functional mobility, and risk of falling. 

Fatigue, Quality of Life, and Overall Effect:Visual Analog Scale (VAS) [54] to evaluate the perceived fatigue after each session; McGill Quality of Life [55] to evaluate the level of patient’s Quality of Life; Global Perceived Effect (GPE) [56], to assess the patient’s subjective perception concerning the impact of intervention on his/her well-being.

Motor evaluations, questionnaires, and statistical analysis were conducted by blinded assessors and statisticians.

### 2.6. Endpoints

The primary endpoint is gait speed measured by the 6-Minute Walking Test. Secondary endpoints include other parameters, such as balance, dynamic balance, risk of falls, and mobility, as well as aspects related to fatigue and quality of life.

### 2.7. Statistics

Absolute values or frequencies were presented for categorical data and mean ± standard deviation for continuous data. The between-group differences in the participants’ characteristics were tested using Pearson’s Chi-squared test for categorical outcome measures and the *t*-test for independent samples or the Mann–Whitney U-test according to the data distribution for continuous outcome measures. The Shapiro–Wilk test was used to verify whether the data were normally distributed or not. 

Linear mixed model analyses for repeated measures were carried out on each outcome measure to examine the efficacy of the novel intervention. The outcome measure was entered as the dependent variable, group and time as fixed effects, and time by group as the interaction effect. The statistical analysis was performed using SPSS Statistics software v.27 (IBM SPSS Inc., Chicago, IL, USA).

## 3. Results

A total of 19 outpatients with Parkinson’s disease (age: 72.6 ± 6.1 years, M/F: 14/5) were included in the study and randomly divided into the experimental group (*n* = 10) and control group (*n* = 9). Groups are comparable at baseline, as reported in Table 1. No significant differences were observed in age, sex, MMSE, and MDS-UPDRS III. 

No side effects were observed during this study phase. Overall, the experimental group reported being more fatigated at the end of the training sessions than the patients in the control group (average of VAS-fatigue over the whole period of training: 6.3 ± 1.8 vs. 4.9 ± 1.1, *p* < value = 0.03). The results of the GPE scale at POST assessment reported that 30% of the participants in the experimental group had a perceived good or very good improvement in their condition, whereas the remaining 70% had a perceived little improvement or any changes in their condition, according to both patients’ and physical therapists’ perspectives. In the control group, 33% of the participants had a perceived good or very good improvement, whereas 67% had a little positive improvement in their condition or no changes, according to both patients’ and physical therapists’ perspectives. No significant between-group differences were noticed in the GPE scale at the end of the intervention (*p*-value = 0.88).

Table 2 summarizes the scores obtained by the primary and secondary outcome measures computed during the motor and functional assessments at PRE, POST, and FU visits. 

A significant group effect was noticed in the total score of the FIM and in its two subscales due to better scores achieved by the control group already at the baseline. The total score of the FIM and its motor subscale showed a significant time effect, indicating an overall improvement over time in both groups. The Mini-BESTest exhibited significant time and group*time effect, indicating that both groups improved over time in this outcome measure, but the increment in the scores was higher in the experimental group. The DGI showed a group*time effect only, indicating that the trends in this outcome were different over time between the two groups. No other significant differences were noticed in the outcome measures computed during the motor and functional evaluation. 

Table 3 shows the scores of the secondary outcome measures related to the quality-of-life assessment. No significant group, time, and time*group effects were found in the McGill QoL subscales related to the physical, physiological, existential, and support domains of the quality of life.

## 4. Discussion

The most notable results of the study primarily concern balance, which improved significantly in the group subjected to a gait training program with sonification and not in the control group. This improvement is evident from the measured changes through the Mini-BESTest and the Dynamic Gait Index. The use of music-based training (pre-recorded stimuli and sonification) may have stimulated an enhancement in the predictability and regularity of movement, assisting patients in better controlling and organizing their movements. The sonification component of the training may have also influenced proprioceptive aspects [41,42,43,44,45,47,51,57], enhancing self-perception of movement through audio-motor feedback [44,47,51]. A similar outcome was observed in a previous study that integrated action observation and sonification into rehabilitation [58].

In general, sonification enhances the explicitness and comprehensibility of motion by describing spatial, temporal, and force-related characteristics through sound [39,40]. Sonification enables the direct and natural perception of one’s own movement and provides effective feedback. Furthermore, the pre-recorded component of the exercises has allowed patients to be stimulated and guided by sound, which, in this case, also incorporates a rhythmic element. It is conceivable that the auditory stimulus constitutes an “added value” in the rehabilitation setting, making the training more engaging and enjoyable compared to movement alone [1].

Contrary to expectations, the primary endpoint of the study (walking speed) did not improve significantly. However, a positive change was observed in the experimental group after training compared to no change in the control group. This endpoint was primarily designed as an indicator for other conditions included in the study protocol, particularly for stroke and multiple sclerosis [10,11,12,14,15,51].

The Timed Up and Go score did not reach statistical significance, although patients in the experimental group improved their scores while maintaining this improvement at the follow-up compared to those in the control group, who remained unchanged. This finding is interesting, considering its implications for balance during the execution of the associated task. Regarding the Functional Independence Measure (especially FIM cognitive), improvements were observed in both groups, making it less likely to be linked to the specificity of the sound stimuli used in this study.

There were no significant results in other secondary outcome measures, such as quality of life (which, in any case, slightly improved in the experimental group) and overall perception of the effect (positive in both groups). The lack of significant results in these outcomes may be attributed to the highly specific nature of the sound stimulus used in the study, which is expected to have a more effective impact on some motor outcomes.

Perceived fatigue was greater in the experimental group, possibly partly due to a higher average age and slightly greater motor functional impairment of the patients (as documented by baseline measurements).

Independent of statistical significance, the data from the study, taking into consideration the small sample size, assume important significance in clinical practice by providing an interesting enrichment of the rehabilitation setting. The sonification approach, in fact, can be considered a low-cost and easily implemented intervention in rehabilitation treatments. This type of sonification can also be used by a physical therapist with the supervision of a music therapist outside the setting. Preliminary data are promising and indicate clear advantages (especially in balance and mobility improvement, and indirectly, in risk of fall reduction) in the use of sonification techniques in gait rehabilitation in Parkinson’s disease.

The study presents some important peculiarities related to the specificity of the sound-musical stimulus used. One of the key features of this training is the type of sonification employed. In contrast to other studies, this approach introduces musical patterns activated by limb movements. 

The patient’s engagement is exclusively motoric, with no cognitive implications.

Another distinctive aspect of this sonification approach is the musical nature of the stimulus, which is not merely a sound but a musical sequence that creates an expected and predictable musical sequence. If the patient’s movement is correct, the musical sequence provides positive feedback, while incorrect movement results in negative feedback through a perceived anomalous musical sequence.

Furthermore, the musical stimulus used consists of a simple succession of sounds, allowing the patient to maintain focus on the movement without distraction from listening. Lastly, another characteristic of this training is the presence of a pre-sonification phase in which pre-recorded musical patterns, supported by a beat, are introduced. 

This is a fundamental point from a rehabilitative perspective, as the opportunity to practice movements through a learning phase guided by rhythmic-musical support potentially facilitates the internalization of the exercise and its execution.

Limitations of the present study are linked to the small sample size. 

Another limitation of the study is the follow-up one month after the end of treatment. A longer follow-up period would be desirable to verify the long-term effect of sonification. However, the characteristics of the sample (outpatients) made it difficult to recall participants back to the hospital after the first follow-up.

Also, technological mediation proved effective but could be enhanced, for example, by increasing the number of sensors to improve movement detection, particularly in more severely affected patients.

From the perspective of stimulus selection, it is possible to enrich the current auditory proposals, for example, in terms of timbre, by creating a musical support with the same musical structure but more aligned with the patients’ musical preferences. This could have a positive impact on non-motor outcomes and improve patient perception of the training.

## 5. Conclusions

Training with sonification (pre-recorded and real-time stimuli) was found to be beneficial, especially for balance, in patients with PD. 

Increasing the sample size and recruiting other patient populations (such as those with stroke and multiple sclerosis) will provide stronger evidence for the obtained results and help define primary outcomes more appropriately for each condition.

Therefore, the formalization of the described training and its implementation in clinical settings and within the research context is desirable through a robust methodological approach and with numerically significant patient samples.

## Figures and Tables

**Table 1 brainsci-13-01586-t001:** Demographic and clinical features of the enrolled patients with Parkinson’s disease.

	Control Group (*n* = 9)	Experimental Group (*n* = 10)	*p*-Value
Age ^§^ [years]	70.0 ± 6.3	74.6 ± 5.6	0.053
Sex * [M/F]	6/3	8/2	0.510
MMSE ^§^ [0–30]	27.7 ± 1.3	27.5 ± 1.3	0.549
MDS-UPDRS III ^§^ [0–132]	14.0 ± 6.2	18.0 ± 6.1	0.182

^§^ Data are presented as means ± standard deviations. * Data are presented as absolute values.

**Table 2 brainsci-13-01586-t002:** Primary and secondary outcome measures computed during the motor and functional evaluations at the baseline (PRE), at the end of the interventions (POST) and at the follow-up visit (FU).

	Control Group ^§^			Experimental Group ^§^			*p*-Value		
	PRE	POST	FU	PRE	POST	FU	G Effect	T Effect	G*T Effect
6MWT [m]	434 ± 82	432 ± 64	441 ± 92	363 ± 91	401 ± 65	394 ± 49	0.21	0.52	0.21
FIM motor [13–91]	82.7 ± 3.0	84.1 ± 2.9	84.4 ± 3.3	77 ± 6.1	79.2 ± 5.5	79.4 ± 5.6	0.02 *	<0.001 *	0.78
FIM cognitive [5–35]	34.3 ± 1.0	34.6 ± 0.5	34.6 ± 0.5	33.6 ± 1.3	33.8 ± 1.3	33.8 ± 1.3	0.01 *	0.80	0.99
FIM total [18–126]	117.1 ± 3.6	118.8 ± 3.2	119.0 ± 3.7	110.6 ± 6.1	113.0 ± 5.4	113.2 ± 5.6	0.01 *	<0.001 *	0.79
Mini-BESTest [0–28]	22.0 ± 2.5	22.6 ± 2.3	22.4 ± 2.4	17.7 ± 3.9	20.6 ± 3.4	21.3 ± 3.0	0.05	0.001 *	0.01 *
DGI [0–24]	21.4 ± 2.2	21.6 ± 1.4	21.4 ± 1.6	17.7 ± 5.0	20.0 ± 3.2	20.8 ± 2.8	0.09	0.09	0.03 *
TUG [s]	9.1 ± 1.9	9.1 ± 2.6	9.1 ± 2.0	12.5 ± 5.7	9.9 ± 1.8	9.9 ± 1.3	0.14	0.16	0.36

^§^ Data are presented as mean ± standard deviation; * *p*-value < 0.05; G: Group; T: Time.

**Table 3 brainsci-13-01586-t003:** Secondary outcome measures related to the quality-of-life assessments performed at the baseline (PRE), at the end of the interventions (POST), and at the follow-up visit (FU).

	Control Group ^§^			Experimental Group ^§^			*p*-Value		
	PRE	POST	FU	PRE	POST	FU	G Effect	T Effect	G*T Effect
McGill QoL— Physical [0–10]	6.2 ± 2.3	5.7 ± 2.1	5.7 ± 2.4	5.1 ± 2.3	4.9 ± 0.9	4.9 ± 1.0	0.23	0.71	0.85
McGill QoL— Psychological [0–10]	7.2 ± 2.3	6.9 ± 1.4	7.0 ± 1.8	5.8 ± 1.9	6.3 ± 1.1	6.7 ± 1.0	0.16	0.67	0.59
McGill QoL— Existential [0–10]	8.0 ± 1.3	7.7 ± 1.3	7.9 ± 1.0	6.6 ± 1.8	7.4 ± 0.9	7.1 ± 1.3	0.07	0.64	0.21
McGill QoL— Support [0–10]	8.5 ± 1.8	8.2 ± 1.6	7.6 ± 1.7	6.8 ± 1.9	6.9 ± 1.3	7.1 ± 1.3	0.07	0.89	0.49

^§^ Data are presented as mean ± standard deviation; G: Group; T: Time; QoL: Quality of Life.

## Data Availability

The data presented in this study are available on request from the corresponding author.

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
