# Peer review of "Movement Sonification Techniques to Improve Balance in Parkinson’s Disease: A Pilot Randomized Controlled Trial"

_brainsci, 2023, doi:10.3390/brainsci13111586_

Round 1

Reviewer 1 Report

Comments and Suggestions for Authors

This is a paper on the effects of sonification techniques on balance ability in Parkinson's patients. I think this is a paper that is convenient because it can be treated using sonification, but will also attract readers' attention.

There are the following fixes:

1. Modification of format and grammar to suit the journal format is required.

2. There are no references required for the introduction. (Lines 35-44) Please provide references for all sentences provided.

3. There are too many paragraphs in the introduction, making it difficult to read. Please shorten the paragraph. Also, please combine similar topics.

4. The P value for age in Table 1 is less than 0.05. Please confirm. Correction using statistics appears to be necessary.

5. Can you do sample size calculation?

6. An explanation of the evaluation tool seems necessary.

7. Were they evaluated by raters blinded to the study?

8. Please fill out the reliability and validity of the evaluation tool.

9. The reasons why sonification techniques improve variables seem necessary to consider. Please provide more details.

Comments on the Quality of English Language

Modification of format and grammar to suit the journal format is required.

Author Response

Reviewer 1

This is a paper on the effects of sonification techniques on balance ability in Parkinson's patients. I think this is a paper that is convenient because it can be treated using sonification but will also attract readers' attention.

There are the following fixes:

  1. Modification of format and grammar to suit the journal format is required.

We have revised the format and grammar, especially in the Introduction and discussion.

  1. There are no references required for the introduction. (Lines 35-44) Please provide references for all sentences provided.

We have added the requested references.

  1. There are too many paragraphs in the introduction, making it difficult to read. Please shorten the paragraph. Also, please combine similar topics.

We have revised the Introduction following your suggestion.

  1. The P value for age in Table 1 is less than 0.05. Please confirm. Correction using statistics appears to be necessary.

We thank the reviewer for the comment, we did not find a between-group significant difference for age, because the p-value for age is not less than 0.05, but equal to 0.053.

  1. Can you do sample size calculation?

A sample size calculation was made for the main study but not for this study that considers only a subgroup of the original RCT (see lines 128-131).

  1. An explanation of the evaluation tool seems necessary.

We have added a short explanation of the well-known evaluation tools (lines 239-260).

  1. Were they evaluated by raters blinded to the study?

The sentence reported in the main text (lines 262-263) clarify this point.

  1. Please fill out the reliability and validity of the evaluation tool.

We have added some references regarding the tools’ description (lines 239-260).

  1. The reasons why sonification techniques improve variables seem necessary to consider. Please provide more details.

We have added in the Discussion section some sentences (lines 337-344) to explain better the reasons why sonification can have improved variables of the study.

Reviewer 2 Report

Comments and Suggestions for Authors

This study aimed to investigate the potential benefits of movement sonification in Parkinson's disease patients. Here, I provide suggestions and questions for enhancing the application of the findings:

1. Title: The title is appropriate.

2. Abstract: The authors aimed to explore the impact of movement sonification on balance. However, it is unclear why they also investigated aspects like fatigue, quality of life, and overall perceived effects. 

3. Abstract: Clarify the meaning of "overall perceived effects" in the abstract.

4. Abstract: Utilizing two groups and three time points (pre, post, and follow-up), a 2-arm ANOVA should have been employed, with a focus on group, time, and the interaction between the two. The statistical test used in the abstract requires clarification.

5. Abstract: The abstract should be rewritten to provide a more balanced summary of the background, results, and conclusion, as it currently appears to be skewed towards the background.

6. Keywords: Although not obligatory, it is recommended to incorporate MeSH (Medical Subject Headings) terms for improved searchability.

7. Introduction: The first paragraph should be accompanied by a reference to support the provided information.

8. Introduction: Avoid the use of excessively short paragraphs for improved readability and coherence. 

9. Introduction: Clarify whether the study's scope includes only PD patients, as suggested in the title, or if it also encompasses "stroke, PD, and multiple sclerosis populations," as stated in the introduction section.

10. Methods: Specify whether patients were assessed during the "on” or “off” phase of their anti-PD medication.

11. Methods: Provide more detail on the randomization process to ensure transparency.

12. Methods: If applicable, include the CONSORT (Consolidated Standards of Reporting Trials) Statement checklist for clinical trials.

13. Methods: Given the study's focus on assessing patients' balance, justify the inclusion of instruments unrelated to balance, such as VAS (Visual Analogue Scale for perceived fatigue after each session), McGill Quality of Life (quality of life), and Global Perceived Effect (GPE).

14. Results: Consider including the Levodopa Equivalent Daily Dose to provide a more comprehensive understanding of the patients' conditions.

15. Results: Explain the rationale behind including the 6MWT, which is not a balance test, in the study.

16. Results: Clarify the purpose of Table 3, especially in the context of the study's objective of assessing participants' balance.

17. Discussion: The discussion section is a crucial aspect of the manuscript and requires more in-depth analysis and exploration of the neurological aspects of the intervention in PD. Avoid excessive repetition and focus on a comprehensive discussion of the results.

18. Discussion: Table 2 indicates that the intervention in both groups was ineffective for FIM cognitive and TUG. These aspects should be thoroughly explored and discussed in this section.

19. Discussion: Address the significant disparity in the number of references between the introduction (49) and the discussion (2). This is the most important aspect why I believe the manuscript should not be approved.

20. Conclusion: The conclusion is appropriate.

21. References: The references are sufficient.

Comments on the Quality of English Language

Several sections of the text require revisions for clarity and improved English.

Author Response

Reviewer 2

This study aimed to investigate the potential benefits of movement sonification in Parkinson's disease patients. Here, I provide suggestions and questions for enhancing the application of the findings:

  1. Title: The title is appropriate.

/

  1. Abstract: The authors aimed to explore the impact of movement sonification on balance. However, it is unclear why they also investigated aspects like fatigue, quality of life, and overall perceived effects.

See the reply to the point 13.

  1. Abstract: Clarify the meaning of "overall perceived effects" in the abstract.

We have clarified the meaning of the words rewriting the text of the Abstract.

  1. Abstract: Utilizing two groups and three time points (pre, post, and follow-up), a 2-arm ANOVA should have been employed, with a focus on group, time, and the interaction between the two. The statistical test used in the abstract requires clarification.

We thank the Reviewer for the comment. Some sentences concerning statistical methods were added in the Abstract (lines 22-25).

  1. Abstract: The abstract should be rewritten to provide a more balanced summary of the background, results, and conclusion, as it currently appears to be skewed towards the background.

We thank the reviewer for the suggestion. The Abstract was rewritten considering the Reviewer’s suggestions. 

  1. Keywords: Although not obligatory, it is recommended to incorporate MeSH (Medical Subject Headings) terms for improved searchability.

We thank the reviewer; we changed the keywords including Mesh terms. The new keywords are Parkinson’s Disease; Neurological rehabilitation; Gait; Postural balance; Sonification; Music therapy.

  1. Introduction: The first paragraph should be accompanied by a reference to support the provided information.

As suggested, we have added the references in the first paragraph of the Introduction.

  1. Introduction: Avoid the use of excessively short paragraphs for improved readability and coherence.

We have substantially revised the Introduction section.

  1. Introduction: Clarify whether the study's scope includes only PD patients, as suggested in the title, or if it also encompasses "stroke, PD, and multiple sclerosis populations," as stated in the introduction section.

We think that the new version of the Introduction and the sentences added to the “Study Design” sub-section (lines 128-131)   can contribute to       clarify this point.

  1. Methods: Specify whether patients were assessed during the "on” or “off” phase of their anti-PD medication.

We thank the reviewer for the comment, we now specify in the text that all patients were assessed during the “on” phase of their anti-PD medication (lines 235-237).

  1. Methods: Provide more detail on the randomization process to ensure transparency.

Participants were randomized to one of the two intervention programs using a list of codes generated through a permuted-block randomization procedure before the beginning of the study. The list was generated by an automatic assignment system, developed in MATLAB, and directly managed by the principal investigator of the study. We added more details in the manuscript as requested (lines 138-141).

  1. Methods: If applicable, include the CONSORT (Consolidated Standards of Reporting Trials) Statement checklist for clinical trials.

We agree with the Reviewer’s suggestion. However, as specified in the text and in our replies to the Reviewers, this paper reports preliminary data referred to a small sub-group taking part of a larger sample of patients with different neurological diseases. The paper containing complete results of the research certainly will follow Consort checklist.

  1. Methods: Given the study's focus on assessing patients' balance, justify the inclusion of instruments unrelated to balance, such as VAS (Visual Analogue Scale for perceived fatigue after each session), McGill Quality of Life (quality of life), and Global Perceived Effect (GPE).

The study mainly focuses on motor outcomes (especially on balance), however, literature reported in the paper showed how   music can also impact on fatigue and quality of life, improving perception of treatment effects. For these reasons we included   other assessment tools, such as the Visual Analogue Scale for perceived fatigue, McGill Quality of Life, and Global Perceived       Effect (GPE).

  1. Results: Consider including the Levodopa Equivalent Daily Dose to provide a more comprehensive understanding of the patients' conditions.

We thank the reviewer for the suggestion. This information could be relevant to characterize participants’ sample. Unfortunately, these data have not been collected during the study because the participants were outpatients not in the hospital's care for drug therapy.

  1. Results: Explain the rationale behind including the 6MWT, which is not a balance test, in the study.

The choice of including the 6MWT as an outcome of the study was made a priori when designing the study, to have a more complete and extensive evaluation of the motor function of the patients, not limited to the balance evaluation. For this reason, the 6MWT was performed in addition to balance tests. Of interest, the main results of the current study on PD patients was reached concerning balance.

  1. Results: Clarify the purpose of Table 3, especially in the context of the study's objective of assessing participants' balance.

See the reply to the point 13.

  1. Discussion: The discussion section is a crucial aspect of the manuscript and requires more in-depth analysis and exploration of the neurological aspects of the intervention in PD. Avoid excessive repetition and focus on a comprehensive discussion of the results.

We have revised the Discussion section adding or modifying in the text some sentences (red colour) focusing on implications in clinical practice.

  1. Discussion: Table 2 indicates that the intervention in both groups was ineffective for FIM cognitive and TUG. These aspects should be thoroughly explored and discussed in this section.

Some sentences commenting on these results were included in the discussion (lines 350-356). However, we think that the result (especially regarding statistical significance) may be related to the smallness of the sample size.

  1. Discussion: Address the significant disparity in the number of references between the introduction (49) and the discussion (2). This is the most important aspect why I believe the manuscript should not be approved.

We thank for the suggestion.  We have added references in the Discussion.

  1. Conclusion: The conclusion is appropriate.

/

  1. References: The references are sufficient.

/

Reviewer 3 Report

Comments and Suggestions for Authors

The manuscript presents a study on the use of music therapy and sonification techniques in neurorehabilitation, more precisely in patients with Parkinson's disease. The aim of the study is to evaluate the effects of applied technique on various motor and functional parameters. In the following, the strengths and weaknesses of the study are highlighted, and suggestions are made for further improvement of the manuscript.

First, I would like to command the authors for the following:

1) Use of an innovative approach, i.e., the study explores a novel approach by combining music therapy and sonification techniques for neurorehabilitation. This approach has the potential to provide a unique and engaging way to improve rehabilitation outcomes,

2) Comprehensive introduction that provides a thorough background on the use of music therapy in rehabilitation and highlights the relevance of the study,

3) Relevance to clinical practice, i.e., a practical clinical concern targeting motor and functional deficits in PD patients. 

However, there are some aspects that need to be improved or properly highlighted in the section Limitations of the study:

1) Small sample size: the sample size of the study is limited to 19 PD patients divided into control and experimental groups. A larger sample size would increase the statistical power of the study and provide more reliable data,

2) The authors stated that this was a single-blinded, parallel-group randomized, controlled trial. Could you please provide more details on the blinding process?

3) Statistical significance of results - the study reports improvements in several outcome measures, some of the statistical significance results are borderline or not significant. It is important to clarify the clinical relevance of these changes and discuss their practical significance

4) Inadequate follow-up: The follow-up period is limited to one-month post-treatment. A longer follow-up period would be valuable to assess the long-term effects of the procedure used. This should at least be highlighted and discussed in Limitations of the study section

5) In general, the Discussion section is relatively brief and could benefit from a more in-depth discussion of the results, their clinical implications, and potential limitations of the study.

6) Ethical Considerations: The manuscript briefly mentions ethical approval and informed consent but does not provide details on the ethical review process and informed consent procedures. Further information on these aspects is needed to ensure the ethical conduct of the study. 

In summary, the manuscript presents an innovative approach to neurorehabilitation through music therapy and sonification techniques. While the study has several strengths, such as clear methodology and relevance to clinical practice, there are areas that need improvement, such as sample size, statistical significance, and follow-up duration. In addition, the discussion section should be more comprehensive.

Comments on the Quality of English Language Some sentences seem to be incomplete, please revise the text again.Some sentences seem to be incomplete, please revise the text again.

Author Response

Reviewer 3

The manuscript presents a study on the use of music therapy and sonification techniques in neurorehabilitation, more precisely in patients with Parkinson's disease. The aim of the study is to evaluate the effects of applied technique on various motor and functional parameters. In the following, the strengths and weaknesses of the study are highlighted, and suggestions are made for further improvement of the manuscript.

First, I would like to command the authors for the following:

1) Use of an innovative approach, i.e., the study explores a novel approach by combining music therapy and sonification techniques for neurorehabilitation. This approach has the potential to provide a unique and engaging way to improve rehabilitation outcomes,

2) Comprehensive introduction that provides a thorough background on the use of music therapy in rehabilitation and highlights the relevance of the study,

3) Relevance to clinical practice, i.e., a practical clinical concern targeting motor and functional deficits in PD patients.

However, there are some aspects that need to be improved or properly highlighted in the section Limitations of the study:

1) Small sample size: the sample size of the study is limited to 19 PD patients divided into control and experimental groups. A larger sample size would increase the statistical power of the study and provide more reliable data,

We totally agree with the Reviewer’s opinion. However, as stated in the main text (lines 152-155), this study is part of a single-blinded parallel-group multicenter randomized controlled trial involving more patients. The study presents only preliminary data concerning a sub-group (from the total sample) of clinically stabilized PD patients (n=19).

2) The authors stated that this was a single-blinded, parallel-group randomized, controlled trial. Could you please provide more details on the blinding process?

New sentences were introduced in the main text to specify the blinding process (lines 262-263).

3) Statistical significance of results - the study reports improvements in several outcome measures, some of the statistical significance results are borderline or not significant. It is important to clarify the clinical relevance of these changes and discuss their practical significance

Clinical relevance of the sonification intervention is underlined in the discussion section (lines 365-373).

4) Inadequate follow-up: The follow-up period is limited to one-month post-treatment. A longer follow-up period would be valuable to assess the long-term effects of the procedure used. This should at least be highlighted and discussed in Limitations of the study section

I agree with the Reviewer’s point of view, however the type of patients recruited in this study makes difficult to do a longer follow-up period. As suggested, we have added a sentence about this in the limitation of the study.

5) In general, the Discussion section is relatively brief and could benefit from a more in-depth discussion of the results, their clinical implications, and potential limitations of the study.

Thank you for your input. We have added in the Discussion section some sentences considering your suggestions.

6) Ethical Considerations: The manuscript briefly mentions ethical approval and informed consent but does not provide details on the ethical review process and informed consent procedures. Further information on these aspects is needed to ensure the ethical conduct of the study.

Ethical details have been included in the main text (lines 132-135) and before references (423-427).

In summary, the manuscript presents an innovative approach to neurorehabilitation through music therapy and sonification techniques. While the study has several strengths, such as clear methodology and relevance to clinical practice, there are areas that need improvement, such as sample size, statistical significance, and follow-up duration. In addition, the discussion section should be more comprehensive.

Thank you for your careful revision. We have completed our revision according to your remarks.

Round 2

Reviewer 2 Report

Comments and Suggestions for Authors

The inclusions made by the authors have improved the text. Although there is still a need for moderate editing of the English language, I recommend the approval of this study.

Comments on the Quality of English Language

Moderate editing of English language required.